# Morphological and Physiological Aspects of Mutable Collagenous Tissue at the Autotomy Plane of the Starfish *Asterias rubens* L. (Echinodermata, Asteroidea): An Echinoderm Paradigm

**DOI:** 10.3390/md21030138

**Published:** 2023-02-22

**Authors:** Iain C. Wilkie, M. Daniela Candia Carnevali

**Affiliations:** 1School of Biodiversity, One Health and Veterinary Medicine, University of Glasgow, Glasgow G12 8QQ, UK; 2Department of Environmental Science and Policy, University of Milan, 20133 Milan, Italy

**Keywords:** advanced glycation end-product, connective tissue, extracellular matrix, interfibrillar crosslink, juxtaligamental cell, ligament, proteoglycan, tendon

## Abstract

The mutable collagenous tissue (MCT) of echinoderms has the capacity to undergo changes in its tensile properties within a timescale of seconds under the control of the nervous system. All echinoderm autotomy (defensive self-detachment) mechanisms depend on the extreme destabilisation of mutable collagenous structures at the plane of separation. This review illustrates the role of MCT in autotomy by bringing together previously published and new information on the basal arm autotomy plane of the starfish *Asterias rubens* L. It focuses on the MCT components of breakage zones in the dorsolateral and ambulacral regions of the body wall, and details data on their structural organisation and physiology. Information is also provided on the extrinsic stomach retractor apparatus whose involvement in autotomy has not been previously recognised. We show that the arm autotomy plane of *A. rubens* is a tractable model system for addressing outstanding problems in MCT biology. It is amenable to in vitro pharmacological investigations using isolated preparations and provides an opportunity for the application of comparative proteomic analysis and other “-omics” methods which are aimed at the molecular profiling of different mechanical states and characterising effector cell functions.

## 1. Introduction

Mutable collagenous tissue (MCT) has the capacity to drastically change its mechanical properties within a timescale of seconds under the control of the nervous system. It is unique to the phylum Echinodermata and is ubiquitous in all five extant echinoderm classes, viz. Asteroidea (starfish), Crinoidea (sea-lilies and featherstars), Echinoidea (sea-urchins), Holothuroidea (sea-cucumbers), and Ophiuroidea (brittlestars) [1].

In all echinoderms, MCT plays a crucial role in two major types of behavioural phenomena. First, by shifting reversibly from a compliant to a stiff state, it serves as an energy-saving mechanism that maintains the posture of the body or its appendages without the need for muscle contraction [2,3,4]. Second, all echinoderm autotomy (defensive self-detachment) mechanisms and asexual reproductive processes depend on the irreversible destabilisation of mutable collagenous structures that traverse the plane of separation [5,6,7,8].

Echinoderm autotomy usually involves the detachment of complex anatomical structures, such as the arms of starfish, featherstars, and brittlestars, and the viscera of featherstars and sea-cucumbers. Echinoderm autotomy planes consequently transect a number of different tissues and organs, which invariably include collagenous components that normally maintain structural and mechanical integrity [6]. These components—ligaments, tendons, and dermal layers—serve the same functions as their counterparts in other phyla: they resist, transmit, and dissipate mechanical forces, and they store and release elastic strain energy. However, they have the additional ability to undergo a rapid loss of tensile strength in response to a range of biotic and abiotic stimuli, which results in the overall mechanical weakening of the autotomy plane and the subsequent breaking away of the anatomical structure that is distal to it under the influence of gravitational, hydrodynamic, or other external forces [6].

The purpose of this review is to illustrate our current level of understanding of the contribution of MCT to echinoderm autotomy mechanisms by bringing together previously published information and new data on the arm autotomy plane of the common North Atlantic starfish *Asterias rubens* L. (order Forcipulatida, family Asteriidae). *A. rubens* is one of the most extensively investigated starfish species. Its capacity for arm autotomy is well known (Figure 1) [9,10,11,12,13,14] and, as autotomy is the main proximate cause of the high damage and regeneration loads sustained by some of its populations [15], this is clearly an important aspect of its life-history and ecology.

After the introduction, we provide an account of the general anatomy of the arm of *A. rubens* and its basal autotomy plane. The review then focuses on the mutable collagenous components of the breakage zones in the dorsolateral and ambulacral regions of the body wall and on the extrinsic stomach retractor apparatus, detailing the available information on, first, their structural organization, and then their physiology. We conclude by showing that the mutable collagenous components of the body wall represent a paradigm in the sense that their properties and functions typify those of MCT at the most investigated echinoderm autotomy planes.

The illustrations are previously unpublished, unless stated otherwise. The abbreviations used in the main text are listed after Section 4.

### Taxonomic Note

Some papers cited herein refer to “*Asterias vulgaris*.” *A. vulgaris* is now regarded as being a junior synonym of *A. rubens* [16]. Mention is also made in the text of *A. forbesi*, which is a sibling species of *A. rubens*. These two species are genetically, although not always morphologically, distinct [17,18].

## 2. Morphological Aspects

### 2.1. Anatomical Background

The following overview is based on several sources [11,19,20,21].

The arms of *A. rubens* consist of a multilayered body wall that encloses a large fluid-filled cavity—the perivisceral coelom. Contained within the arms and attached to the internal surface of the body wall are the gonads and components of the digestive system—the pyloric caeca and the extrinsic elements of the stomach retractor apparatus (Figure 2 and Figure 3A).

The body wall has three main layers: an outer epidermis of ectodermal origin, whose apical surface is in contact with the external environment, a middle layer of mesodermal origin known as the dermis (although the appropriateness of this terminology has been questioned [22]), and an inner coelothelium (coelomic epithelium), also mesodermally derived, whose apical surface is in contact with the perivisceral coelomic fluid. This basic organisation is complicated by the presence of spines, by papulae and pedicellariae projecting from the dorsal and lateral regions of the body wall, and by tube-feet in the ambulacral (ventral) region. In addition, as well as incorporating the radial nerve cord, which is continuous with the epidermis [23], the ambulacral body wall houses a number of longitudinal vessels and coelomic compartments, including the radial water vascular canal (which is a component of the hydraulic system of the tube-feet), the radial haemal sinus (which is involved in nutrient translocation [24]), and the radial perihaemal (or hyponeural) canal (Figure 2).

The endoskeleton of the body wall is located in the dermis and takes the form of ossicles composed of a three-dimensional, mainly calcitic, network (stereom) that are connected at mobile articulations by interossicular ligaments and muscles. In the dorsal and lateral regions of the arm body wall, the ossicles form a regular mesh linked to a single longitudinal row of carinal ossicles at the dorsal midline. The endoskeleton of the ambulacral body wall is a longitudinal series of paired rafter-like ambulacral ossicles that are inclined towards the mouth and have smaller adambulacral ossicles attached to their outer edge (Figure 2).

In *A. rubens* and other asteriid starfish, autotomy always involves detachment of the arm at a transverse level close to its base (Figure 1). This is in contrast to the capacity of certain other starfish, notably members of the family Luidiidae, to discard arms at multiple transverse levels [6,25]. In light of the information outlined above, it is obvious that the basal arm autotomy plane of *A. rubens* transects several different macro- and micro-anatomical components, each of which undergoes rupture during arm detachment, viz. the body wall including the longitudinal vessels and coelomic compartments contained within it, the radial nerve cord, the common pyloric duct, which connects the pyloric region of the stomach to the paired pyloric caeca, and the extrinsic stomach retractor apparatus, which links the cardiac region of the stomach to the dorsal surface of the ambulacral body wall distal to the autotomy region. It should be noted that the gonads and the gonoducts through which gametes are delivered into the external environment are located distal to the autotomy region and undergo no breakage process at autotomy (Figure 2 and Figure 3A).

The term “breakage zone” is used herein to denote the level of rupture in an individual microanatomical component. In the autotomy planes of all animal phyla, breakage zones are often localised and predictable due to the presence of adaptations that result in a relatively simple break at autotomy, which facilitates subsequent wound-healing and regeneration. Such breakage zones are usually either permanent, preformed planes of mechanical weakness, or potential planes of mechanical weakness that are destabilised by a physiological process only when autotomy has been elicited [6]. This review focuses on the involvement of MCT at breakage zones of the latter type in the body wall and extrinsic stomach retractor apparatus of *A. rubens*. After autotomy, the ruptured ends of the pyloric duct have a ragged and variable appearance (Figure 3A). This observation, together with the absence of externally visible indications (Figure 3B) or histological evidence (I.C. Wilkie and M.D. Candia Carnevali, unpublished data) of a localised breakage zone in the intact pyloric duct, suggests that its fracture may be facilitated only by the inherently low resistance of the duct epithelium to uniaxial tension.

In this review, the term “dorsolateral” refers to the combined dorsal and lateral regions of the body wall. Because of their different microanatomical organisations, the dorsolateral and ambulacral regions are dealt with in separate sections below.

### 2.2. Dorsolateral Body Wall

#### 2.2.1. Organisation of Intact Dorsolateral Body Wall

The general histological and ultrastructural organisation of the dorsolateral body wall of *A. rubens* was described by Wilkie et al. [19]. More specific aspects of its morphology were addressed by Blowes et al. [20], Schwertmann et al. [21], and Wilkie and Candia Carnevali [26]. In this section, information derived from Wilkie et al. [19] is illustrated mainly with previously unpublished micrographs and combined with new data on proteoglycan histochemistry.

The epidermis comprises an outer extracellular surface coat [27]; a monolayered, primarily columnar epithelium that includes support cells, secretory cells, and sensory cells, the basal portions of which are entwined by a basiepithelial neural plexus; an inner basal lamina (=lamina lucida + lamina densa). At autotomy, the epidermis splits along a transverse groove (Figure 4A,E) that is sometimes visible in small living animals. There appear to be no modifications of the epidermis at the breakage zone, other than a reduction in its thickness.

The coelothelium is a pseudostratified myoepithelium with both apical peritoneocytes and subapical, longitudinally orientated myocytes attached to a basal lamina. On the dorsal side of this basal lamina there are several tissue layers that include circumferentially orientated myocytes and have been interpreted as belonging to the dermis [28], although in *A. rubens* they are separated from the densely fibrous layer of the inner dermis (see below) by a space delimited by a thin layer of squamous cells (Figure 4B,C). In the midline of the dorsalmost body wall, the longitudinal myocyte layer is greatly expanded to form the apical muscle (Figure 2 and Figure 4G). Also extending from the coelothelium are elements of a microcanalicular system that extends into the dermis, the function of which is unknown [29,30]. No morphologically distinct breakage zone has been detected in the apical muscle (Figure 4G) or any of the other tissue layers that are ventral to the dense inner dermis (I.C. Wilkie and M.D. Candia Carnevali, unpublished data).

**Figure 4 marinedrugs-21-00138-f004:**
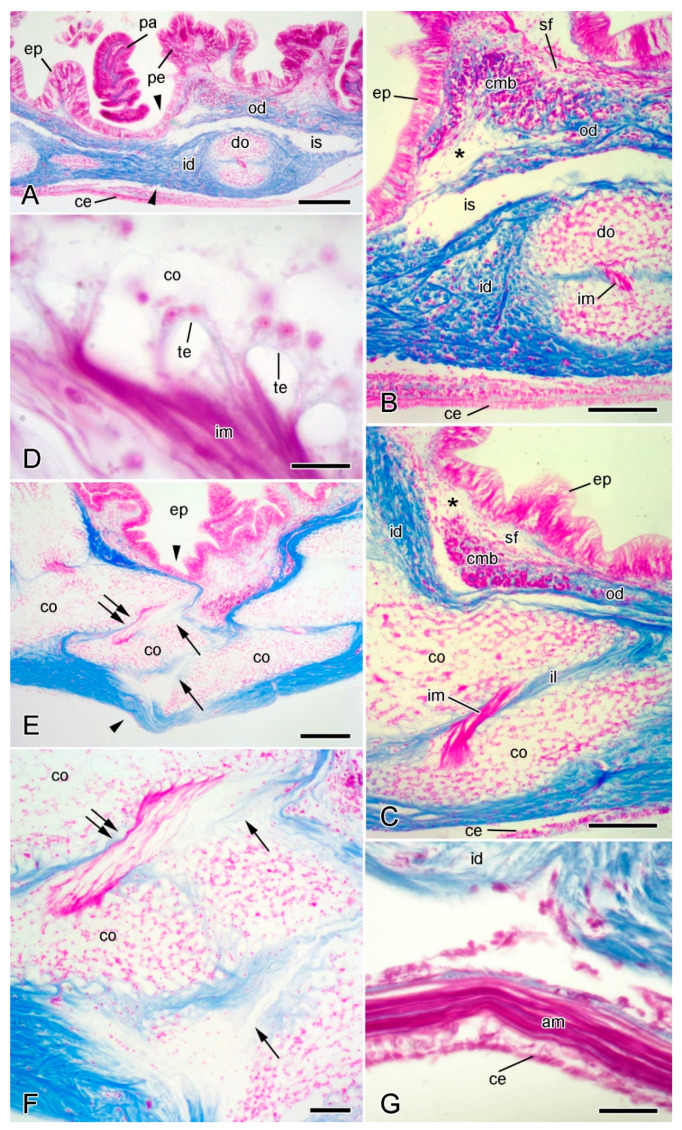
Light microscopy (LM). Histology of dorsolateral body wall. Parasagittal (**A**,**B**) and sagittal (**C**–**F**) sections stained with Milligan’s trichrome [31]: collagenous tissue is blue, muscle and other cellular tissues are magenta. In all sections, distal end is on left and dorsal side is on top. (**A**) General view including intact breakage zone (approximate position indicated by arrowheads). Scalebar = 0.2 mm. (**B**) Intact breakage zone. Scalebar = 0.1 mm. (**C**) Breakage zone at level of carinal ossicles. Scalebar = 0.1 mm. (**D**) Attachment area between interossicular muscle and carinal ossicle. Scalebar = 10 µm. (**E**–**G**) Autotomising breakage zone (between arrowheads). (**E**) General view (coelomic layers missing). Double arrows indicate rupturing intercarinal muscle and single arrows rupturing intercarinal ligament. Scalebar = 0.2 mm. (**F**) Rupturing intercarinal muscle (double arrows) and intercarinal ligament (single arrows). Scalebar = 50 µm. (**G**) Coelomic side of autotomising body wall. Although inner dermis is disintegrating, adjacent apical muscle and coelothelium are intact. Scalebar = 20 µm. Asterisk, expanded sparsely fibrous region; am, apical muscle; ce, coelothelium; cmb, circular muscle bundle; co, carinal ossicle; do, dorsolateral ossicle; ep, epidermis; id, inner dermis; il, interossicular ligament; im, interossicular muscle; is, intradermal space; od, outer dermis; pa, papula; pe, pedicellaria; sf, sparsely fibrous region; te, tendon.

The dermis consists mainly of fibrous collagenous tissue. In the regions of the body wall between the papulae, pedicellariae, and spines, the dermis is divided into two main horizontal layers—the outer and inner dermis—by a space of variable width that is lined with squamous cells (Figure 4A–C). The outer dermis has a sparsely fibrous region which is adjacent to the epidermis in which the most abundant cellular constituents are fine muscle fibres of varying orientation. This is continuous with a deeper sublayer of densely packed collagen fibres that lacks myocytes (Figure 4A–C). The inner dermis is generally much thicker than the outer dermis and consists of densely packed collagen fibres and endoskeletal ossicles. The ossicles are surrounded by collagen fibres and almost the whole surface of each ossicle is linked to the dermal collagenous tissue by fibres that penetrate and branch within stereom spaces [20] (Figure 4B,C). The collagenous envelope is interrupted only at the attachment areas of the small muscles that link adjacent ossicles and that are attached to the stereom by tendon loops (Figure 4C,D).

At autotomy, the dermis ruptures at a level just distal to a prominent transverse row of dorsolateral ossicles (Figure 3B). The dermal breakage zone is distinguished by two features: (1) there is an expanded sparsely fibrous region (ESFR) that extends for most of the thickness of the outer dermis; (2) immediately proximal to the ESFR there is a thick circumferential band of circular muscle consisting of cylindrical bundles of myocytes (Figure 4B,C). Neither of these features is present at any other level in the arm.

In ultrathin sections prepared for transmission electron microscopy, the extracellular components of the ESFR consist mainly of an irregular network of granulo-filamentous material that varies in density and within which there are widely separated bundles of cross-banded collagen fibrils and scattered microfibrils ca. 10 nm in diameter (Figure 5A,B). The granulo-filamentous network is linked to the basal lamina of the epidermis (Figure 5B). In the inner dermis, bundles of collagen fibrils form a dense meshwork and are surrounded by granulo-filamentous material and clusters of ca. 10 nm microfibrils (Figure 5C,D). The tendons of the interossicular muscles extend from the basal lamina of the muscle cells and consist of parallel aggregations of fine filaments that are directly attached to the stereom trabecular coat (Figure 5E,F) [32].

In view of the possible contribution of proteoglycans (PGs) to the mechanical properties of collagenous tissue, including MCT [1,33,34,35], the disposition of PGs in the dermis of *A. rubens* has been investigated (I.C. Wilkie and M.D. Candia Carnevali, unpubl. data) by TEM histochemical methods that employ the cationic staining agents polyethyleneimine (PEI) [36] and cupromeronic blue (CMB) [37]. Both markers indicated that PGs are widely distributed in the dermis. In the ESFR of the outer dermis, and in the inner dermis, they are visualised as electron-dense granules attached to the surface of collagen fibrils at the same position in each D-period, and they also appear to form interfibrillar connections (Figure 6A,B,D,F,G), as has been observed in other echinoderm mutable collagenous structures [37,38,39,40]. In addition, CMB, although not PEI, staining indicates that PGs are a significant component of the granulo-filamentous material of the ESFR (Figure 6C,E).

Regarding the cellular components of the dermis, of particular interest are the juxtaligamental cells, which are implicated in the control of MCT tensile properties and which are recognised by their large (diameter >100 nm) intracellular dense-core vesicles (LDCVs) [26]. The ESFR includes very sparsely distributed LDCV-containing cell processes (Figure 5A). These are of two types, distinguished by their LDCV size (maximum diameters are 670 nm and 260 nm, respectively), and occur singly or in small heterotypic clusters. They are not separated from the extracellular matrix by a basal lamina.

In the inner dermis there are small clusters of LDCV-containing processes alone as well as linear aggregations of cell bodies and processes. These belong to three cell-types: types 1 and 2 contain LDCVs resembling the larger and smaller LDCVs of the ESFR (maximum diameters are 700 nm and 350 nm, respectively), and type 3 has LDCVs that are up to 150 nm in diameter. In aggregations that include cell bodies, the cell bodies form an outer layer surrounding a bundle of largely parallel type 3 processes (Figure 7A–C). Most cell bodies contain type 1 LDCVs; somata containing type 3 LDCVs are present albeit rare; and somata with type 2 LDCVs were not observed. Also present are cell bodies containing many spherical inclusions with medium electron opacity, a maximum diameter of 800 nm, and an inconsistently visible boundary layer (Figure 7B). These look like typical lipid droplets [41] and resemble the lipid inclusions occurring in neuronal perikarya in the Lange’s nerves of *A. rubens*, which are part of the hyponeural motor system [42]. Aggregations that include cell bodies have an outermost layer consisting of sparse agranular somata with a spindle-shaped profile and very thin extensions that form a loose, discontinuous sheath (Figure 7B,C). These cells are ciliated; their cytoplasm includes mitochondria and vesicles that are of variable size, have heterogeneous contents, and, in some cases, appear to be undergoing exocytosis. Their cytological features indicate that these cells may be gliocytes, i.e., cells that regulate the perineuronal environment [43]. No basal lamina was observed between any of these cellular components and the dermal extracellular matrix.

In preparations stained with CMB (however, not in those stained with PEI), elongated electron-dense deposits are present at the membrane of type 1 LDCVs in the ESFR and inner dermis (Figure 7D–F). Some are in the form of paired deposits on either side of the LDCV membrane (Figure 7E), suggesting the presence of PG-containing transmembrane complexes (perhaps related to the PG-containing SV2 membrane transporter in the secretory vesicles of all vertebrate neurons and endocrine cells [44,45]). Welsch et al. [46] observed similar CMB-positive precipitates on the surface of juxtaligamental LDCVs in a featherstar (class Crinoidea). As PGs are present in the cytoplasmic granules of a wide range of neuronal and other cell types that have a secretory function [47], their occurrence in juxtaligamental LDCVs corroborates the postulated secretory role of JLCs.

#### 2.2.2. Organisation of Autotomising Dorsolateral Body Wall

Wilkie et al. [19] described the changes in the organisation of the dorsolateral body wall of animals undergoing autotomy induced by the intracoelomic injection of 0.56 M KCl.

Immediately before autotomy, the dorsolateral body wall constricts deeply at the breakage plane (Figure 1B and Figure 4E). The resulting depression coincides with the position of the thick band of circular muscle (Figure 4E). Therefore, the latter appears to function as a tourniquet that constricts the body wall before autotomy and thus facilitates rapid wound closure after autotomy.

The outer dermis breaks at the level of the ESFR and it is hypothesised that, in view of its thinly scattered content of collagen fibres (= fibril bundles), the ESFR is a preformed zone of weakness in the outer dermis, whose function is to determine the line of rupture of the epidermis.

During autotomy, the inner dermis becomes highly disorganised, with disarrangement and disaggregation of the collagen fibres. This disintegration is limited to the autotomy plane and is apparent in the ligaments at the junctions between dorsolateral ossicles, such as those of the carinal series (Figure 4E,F). Disintegration is not a result of externally imposed force, as can be seen from sections in which, despite breakdown of the inner dermis, both the epidermis and coelothelial layers are intact (Figure 4E,G). Rupture of the inner dermis is thus achieved by an endogenous mechanism resulting in the loss of tensile strength of the whole body wall. There are no ultrastructural indications that individual collagen fibrils undergo disaggregation, which has been proposed as the possible basis of the mutability phenomenon in featherstar ligaments [37]. Dermal disarrangement is thus likely to result from the loss of cohesion between collagen fibrils or fibres.

Autotomy also necessitates the rupture of interossicular muscles at the breakage plane. Fracture occurs within the body of the muscle and not at the level of the tendons (Figure 4E,F).

Several changes are seen in type 1 LDCV-containing juxtaligamental cellular elements of the autotomising inner dermis. The most consistent change is a reduction in the electron density of type 1 LDCV contents. Also, type 1 cells are more difficult to find, there being loose clusters of type 3 processes which are uncharacteristically lacking the associated type 1 cell bodies or processes. No equivalent changes were discernible in type 2 or type 3 LDCV-containing processes, although there may be a reduction in the granule content of the latter (Figure 7G,H).

### 2.3. Ambulacral Body Wall

#### 2.3.1. Organisation of Intact Ambulacral Body Wall

The organisation of the ambulacral body wall of *A. rubens* (Figure 2 and Figure 8A–C) has not been previously described in detail, although Blowes et al. [20] and Schwertmann et al. [21] provided comprehensive information on its endoskeleton. The following account is based on the present authors’ investigations (I.C. Wilkie and M.D. Candia Carnevali, unpublished data).

The epidermis that is adjacent to the adambulacral ossicles resembles that of the dorsolateral body wall, whereas, on the adradial side of the tube-feet, i.e., at the apex of the ambulacral groove, it takes the form of a pseudostratified neuroepithelium that constitutes the radial nerve cord [23,48]. The coelothelium of the ambulacral body wall is a simple cuboidal epithelium. The dermis is occupied mostly by a longitudinal series of ambulacral and adambulacral ossicles, each of which, as in the dorsolateral body wall, is completely surrounded by, and connected to, fibrous collagenous tissue, except at the attachment areas of interossicular muscles. Collagenous tissue is thus present both as interossicular ligaments and as a continuous inner dermal sheath on the dorsal and ventral sides of the ossicles (Figure 8B,C and Figure 9A,B,E).

Adjacent pairs of ambulacral ossicles and their attached adambulacral ossicles are connected at mobile joints by longitudinal ligaments and by longitudinal interambulacral and interadambulacral muscles. Autotomy involves the rupture of these longitudinal ligaments and muscles at the base of the arm (Figure 9A,E).

The histological and ultrastructural organisation of the longitudinal interambulacral ligaments is illustrated in Figure 9 and Figure 10. The ligaments consist of a dense array of collagen fibres which are orientated mainly diagonally with respect to the distal and proximal surfaces of the opposing ambulacral ossicles. There are also abundant cell bodies and processes that have a strong affinity for acidic histological stains (Figure 8C and Figure 9B,C).

The fibres consist of parallel assemblies of cross-banded collagen fibrils (Figure 10A). The cellular constituents include two types of LDCV-containing cell bodies and processes, one type in which LDCVs usually have a circular profile and diameter up to 540 nm, the other in which LDCVs have a circular or capsule-shaped profile up to 400 nm long. The processes always occur in heterotypic clusters of two or more. The LDCV-containing cell bodies form aggregations together with bundles of processes; however, these are less compact and less regularly organised than those of the dorsolateral dermis: somata and processes containing the larger type of LDCV do not form an outer layer (Figure 10B–D). Also present are somata with possible lipid droplets and agranular processes containing microtubules and small electron-lucent vesicles that are up to 140 nm in diameter. At the outer margin of these aggregations, as in those of the dorsolateral dermis, there are glia-like agranular somata whose cytoplasm contains heterogeneous vesicles, rough endoplasmic reticulum, and a Golgi complex (Figure 10D). The processes of these cells extend along the aggregations; however, they do not form a continuous sheath. There is no basal lamina between the cellular components and the extracellular matrix.

As for the interossicular muscles of the dorsolateral body wall, the longitudinal interambulacral muscles are attached to the ambulacral ossicles by tendons which consist of extensions of the basal lamina of the muscle cells [49,50].

A preliminary examination of the histological organisation of the radial nerve cord, radial haemal sinus, and radial water vascular canal in the basal breakage region revealed no evidence for the presence of a morphologically differentiated breakage zone in any of these structures.

#### 2.3.2. Organisation of Autotomising Ambulacral Body Wall

It has been reported that, at autotomy, the ambulacral body wall of *A. rubens* usually breaks at the level of the joint between either the third and fourth or fourth and fifth pair of ambulacral ossicles (Figure 8A,B) [10,12,51]. While this may apply to larger animals, the present authors observed that, in smaller animals (centre to arm-tip radius of 24–36 mm), most breakages occurred at the joint between the second and third ambulacral pair (Figure 9E). However, it is clear that, rather than there being a specific interambulacral breakage zone at the base of the arm, there is a breakage region comprising two or more interambulacral joints. Similar diffuse breakage regions encompassing more than one interambulacral joint have been identified in other asteriid species [52,53,54,55].

At the joint undergoing disarticulation, the fibres of the longitudinal interambulacral ligament and the inner dermal sheath disaggregate into wispy strands (Figure 9A,D–F), as observed in the dorsolateral dermis. The longitudinal interambulacral muscles at the breakage level rupture through the middle, leaving remnants on both the retained and discarded wound surfaces (Figure 9E,F). Muscle rupture is not limited to the breakage level and can be seen in interambulacral joints that are proximal and distal to the disarticulating joint (Figure 9E). Figure 9G shows a muscle rupturing at a joint where the adjacent interambulacral ligament, inner dermal sheath, and coelothelium appear to be intact, suggesting that muscle rupture is achieved by an endogenous mechanism which is specific to the muscles and is not the result of externally applied tensile force.

### 2.4. Extrinsic Stomach Retractor Apparatus

The available information on the morphology and composition of the extrinsic stomach retractor apparatus (ESRA) of *A. rubens* and its close relative *A. forbesi* is limited to its histological organisation and histochemical properties [56,57,58]. Anderson [56] also provided a detailed historical account of views on its structure and function.

The ESRA consists of five pairs of triangular sheets, with each pair extending from the cardiac stomach to the dorsal side of the ambulacral body wall (Figure 8D,E). The two upper edges of each triangular sheet are thickened to form the proximal and distal retractor strands (Figure 2 and Figure 11A–C). At their dorsal ends, the four retractor strands of each arm are connected to a band of tissue which is referred to as a “nodule” [56] that is attached to the visceral side of the stomach wall (Figure 8E).

The retractor strands contain longitudinally orientated collagen fibres and muscle fibres, and have an outer layer of squamous coelothelium. The collagen fibres tend to be more abundant at the adradial side of each strand and form partitions that delimit longitudinal compartments containing bundles of muscle fibres (Figure 11B–E).

The mesentery-like sheet of tissue that extends from the retractor strands to the ambulacral body wall consists of a central layer of collagen fibres sandwiched between adradial and abradial coelothelia. It appears to lack muscle fibres (Figure 11D,F).

The collagen fibres of the retractor strands and mesentery merge with the inner dermal sheath on the dorsal side of the ambulacral ossicles (Figure 11A,B).

At autotomy, the retractor strands and mesentery break away at their junction with the nodule (Figure 8E and Figure 12) (I.C. Wilkie and M.D. Candia Carnevali, unpublished data).

### 2.5. Comments on Morphological Aspects

#### 2.5.1. Morphological Insights into Breakage Mechanisms

The breakage zone of the dorsolateral body wall is characterised by the presence of an expanded, sparsely fibrous region (ESFR) in the outer dermis. It is hypothesised that the ESFR is a preformed zone of intrinsic mechanical weakness that determines the line of rupture of the overlying epidermis. In contrast to this, it is apparent from the histological sections of autotomising arms that, within the densely fibrous inner dermis of the dorsolateral body wall and the longitudinal ligaments and inner sheath of the ambulacral body wall, there is a potential zone of inducible mechanical weakness at which these tissues can undergo disaggregation by an endogenous process, resulting in their complete rupture and, since they are the dominant structural components of the body wall, detachment of the whole arm. Histological observations thus provide evidence that the densely fibrous dermal layers at the level of the autotomy plane are composed of mutable collagenous tissue (MCT).

There is, at present, insufficient information on the ESRA to justify any opinion on the nature of its collagenous components or the process by which it separates from the cardiac stomach during arm detachment.

At autotomy, interossicular muscles in the dorsolateral body wall and longitudinal interambulacral muscles in the ambulacral body wall rupture through their mid-substance. Such autotomy-associated intramuscular rupture, which is accomplished by means of an unidentified endogenous mechanism, also occurs in the asteriid *Pycnopodia helianthoides* [55] and in species belonging to other orders: *Luidia ciliaris*, *Astropecten irregularis* (both order Paxillosida), and *Henricia oculata* (order Spinulosida) (I.C. Wilkie and F. Oladosu, unpublished data). This contrasts with the pattern of muscle breakage in autotomising brittlestar arms, in which intervertebral muscles at the affected intersegmental arm-joint detach at the level of the tendons linking them to a vertebral ossicle. These tendons are mutable collagenous structures that are destabilised at autotomy and are in close contact with juxtaligamental cell processes [6,60]. Although starfish tendons resemble those of brittlestars in consisting of loop-like extensions of the basal lamina of the muscle cells [49,50,60], they lack closely associated juxtaligamental components and, therefore, are unlikely to show mechanical mutability.

There is no morphological evidence for the presence of differentiated breakage zones, or any other autotomy-related modifications, in the pyloric duct, apical muscle, radial water vascular canal, radial haemal sinus, or radial nerve cord of *A. rubens*. This may be related, at least partly, to the inherently low tensile strength of these components and the consequent absence of evolutionary pressure favouring structural adaptations that localise or facilitate fracture. However, the occurrence of autotomy-related features in the brittlestar radial water vascular canal, radial haemal sinus, and radial nerve cord [61] suggests that a more thorough comparative investigation of starfish autotomy planes might be enlightening.

#### 2.5.2. Juxtaligamental Components and Their Significance

The presence of juxtaligamental cells (JLCs) in the dermal layers of the dorsolateral and ambulacral body wall is further evidence that these consist of MCT. All mutable collagenous structures in the five echinoderm classes are permeated by, or closely adjacent to, juxtaligamental cell processes, and all non-mutable collagenous structures of echinoderms that have been investigated lack these [26].

As in most other mutable collagenous structures, the dermis of *A. rubens* contains different JLC types, which are distinguishable by the size of their LDCVs. Of the three types that can be identified in the dorsolateral dermis, those with smaller LDCVs—called types 2 and 3 herein—may be regular aminergic or peptidergic neurons whose main function is signal transmission [46], whereas those with very large LDCVs—type 1—are more likely to be non-regular neurons whose main function is secretion. Type 3 JLCs were not found in the longitudinal interambulacral ligaments, although cells with 140 nm electron-lucent vesicles were present.

In both the dorsolateral inner dermis and longitudinal interambulacral ligaments, juxtaligamental somata occur with other cell types in small aggregations that resemble micro-ganglia in having centrally located bundles of cell processes and a loose outer layer of glia-like cells. These aggregations share morphological features with Lange’s nerve—the hyponeural component of the radial nerve cord [42]: they contain neuronal perikarya with lipid inclusions, cells with lysosome-like vesicles (called “macrophage-like cells” by von Hehn [42]), and cell bodies and processes with LDCVs that have a diameter of 100–450 nm (called “neurosecretory cells” by von Hehn [42]). Wilkie and Candia Carnevali [26] conjectured that each of these aggregations represents a terminal hyponeural branch and its target cells, the latter being type 1 JLCs. Cellular aggregations containing juxtaligamental somata and having ganglion-like features are associated with the mutable collagenous structures of the other echinoderm classes, with the possible exception of the Holothuroidea [26].

The JLCs in the dorsolateral inner dermis exhibited ultrastructural changes at autotomy. Various modifications in the ultrastructural cytology of JLCs have been observed in a range of mutable collagenous structures undergoing autotomy-related destabilisation [26]. However, there is uncertainty about whether such changes are incidental consequences of cell damage caused by tissue disruption or evidence for JLC involvement in the destabilisation mechanism.

## 3. Physiological Aspects

### 3.1. Dorsolateral Body Wall

Verification that the body wall dermis of *A. rubens* consists of MCT was obtained from in vitro experiments using isolated tissue preparations. Wilkie et al. [19] compared the mechanical behaviour of longitudinal strips of dorsolateral body wall that were traversed by the autotomy plane (ABW preparations) with that of strips excised from more distal regions of the arm (DBW preparations). In creep tests, in which the body wall samples were subjected to a constant tensile load of 5 g (Figure 13A–C and Figure 14), stimulation of the ABW preparations with sea-water containing 100 mM K^+^ ions was followed by an abrupt increase in their extension rate, culminating in rupture at the autotomy plane (Figure 14B). In effect, 100 mM K^+^ ions caused a reduction in ABW viscosity which mimicked events at autotomy. DBW preparations were much less responsive to high [K^+^]: this had no effect or caused an increase in extension rate that was either transient or persistent but slight in magnitude (Figure 14B); no DBW preparations ruptured.

Anaesthetisation by sea-water containing 0.1% propylene phenoxetol, an agent that blocks conduction in squid and brittlestar axons, completely abolished the responses of ABW and DBW preparations to high [K^+^], indicating that the action of K^+^ ions is nervously mediated.

Wilkie et al. [19] investigated the pharmacology of ABW preparations (Table 1). Acetylcholine (10^−4^ M) caused an abrupt reduction in the extension rate and stopped further lengthening (Figure 14C). ABW preparations include longitudinally orientated myocytes in the outer dermis and coelothelium (see Section 2.2.1 above). However, of hundreds of preparations treated with acetylcholine, as well as the cholinergic agonists methacholine and carbachol (see below), only three showed a transient phase of shortening attributable to active contraction (Figure 14C), suggesting that the more usual sustained arrest of extension caused by acetylcholine represents stiffening of the dermal MCT rather than myocyte contraction. A clearer indication of the importance of the MCT was provided by an ABW preparation in which 10^−4^ M acetylcholine caused the complete arrest of extension under a high load of 13.6 g. It was calculated that, had they been responsible for this reaction, the longitudinally orientated myocytes would have had to generate a tension of 4.4 MPa, which is well above the maximal tension which was recorded in any muscle (1.5 MPa: anterior byssus retractor of *Mytilus* [62]). The usual response to cholinergic agonists must involve MCT stiffening, although the possibility that the isometric contraction of muscular elements makes a contribution cannot be discounted.

Tests employing a range of amines and cholinergic agonists and antagonists (Table 1) indicated that the “stiffening” action of acetylcholine may be mediated by muscarinic (or muscarinic-like) receptors: (1) of the cholinergic agonists which were tested, only the muscarinic agents methacholine and carbachol had a consistent stiffening effect, with the former having a greater effect than the latter; (2) of the cholinergic antagonists which were tested, only the muscarinic antagonist atropine blocked the action of acetylcholine. Atropine (10^−3^ M) by itself caused an increase in extension rate (i.e., “destiffening”) (Figure 14D). None of the other agents which were tested had a significant destiffening effect or blocked K-induced destiffening.

Marrs et al. [15] discovered that there are size-related trends in the mechanical properties of ABW and DBW preparations from *A. rubens*. Employing force-extension tests, in which the preparations were stretched at a predetermined rate (Figure 13D,E), they found that the yield stress, ultimate stress, and Young’s modulus showed a positive correlation with size (Figure 15), i.e., the tensile strength and stiffness of the dorsolateral body wall were greater in larger individuals. In the same investigation, Marrs et al. [15] demonstrated that size was correlated negatively with the incidence of arm damage attributable to autotomy in several populations of *A. rubens* and positively with the autotomy response time (the delay between the onset of stimulation and arm detachment). It was hypothesised that, in larger starfish, increased tensile strength and stiffness conferred greater mechanical resistance to predator-induced damage, which reduced the usefulness of autotomy as a defence mechanism, and that this had resulted in a decrease in the incidence of autotomy and, over an evolutionary timescale, an increase in the autotomy response time through a relaxation of selection pressure because of the reduced fitness benefit of autotomy.

Since size is an unreliable indicator of age in *A. rubens* [63], the trends in body wall mechanics identified by Marrs et al. [15] can be interpreted only tentatively as an age-related phenomenon. However, they accord with the age-related increase in stiffness and tensile strength demonstrated by some mammalian collagenous structures [64,65,66]. One factor that may contribute to these changes in mammalian collagenous tissue is the progressive accumulation of advanced glycation end-products (AGEs) that form intermolecular crosslinks at different structural levels within the extracellular matrix [67,68]. The incubation of dorsolateral body wall samples from *A. rubens* in a solution of the reducing sugar ribose, a method used to induce AGE formation in model mammalian systems [67,69], resulted in significant changes in certain mechanical parameters (Figure 16), indicating that the non-enzymatic glycation of collagenous tissue may also operate as an endogenous ageing mechanism in echinoderms (I.C. Wilkie and J.J. Keane, unpubl. data).

### 3.2. Ambulacral Body Wall

The physiology of the ambulacral body wall has been investigated using isolated preparations of the ambulacral ridge, each of which includes a segment of the mouth frame and a breakage region (Figure 8A) (I.C. Wilkie and G.V.R. Griffiths, unpublished data).

The mechanical responses of these preparations to neuroactive agents are more varied and more difficult to interpret than those of the dorsolateral preparations, due to the greater contribution of the contractile components in the former, i.e., the longitudinal interambulacral muscles. The range of responses to sea-water containing an elevated K^+^ concentration are illustrated in Figure 17A. In most preparations, high [K^+^] caused an increase in the extension rate that, in around half of these cases, led to breakage—always at the autotomy region. The response to high [K^+^] was dose-related, with the maximum effect occurring at 100 mM K^+^ (Figure 17B). High [K^+^] invoked shortening in a few preparations. This effect was usually transient and, in some preparations, was followed by an increase in the extension rate and breakage at the autotomy region (Figure 17Ad).

High [K^+^] thus produced responses in two effector systems: contraction of the longitudinal interambulacral muscles and destiffening of mutable collagenous structures in the autotomy region.

The preparations treated with acetylcholine never broke and, in a dose-response experiment, showed a pattern of reactions that could be interpreted as resulting from a concentration-related increase in the force of contraction of the longitudinal interambulacral muscles (Figure 17C). While acetylcholine-induced shortening of preparations is attributable to muscle contraction alone, other effects that were observed, i.e., decreased extension rate and arrest of extension, could have been due wholly or partly to MCT stiffening. Evidence for the involvement of MCT was obtained in a preliminary pharmacological investigation (Table 2), which revealed that atropine by itself increased the extension rate of these preparations, as was also the case for the dorsolateral body wall preparations of *A. rubens* (see Section 3.1. above) and ambulacral preparations of *Pycnopodia helianthoides* [55]. This raises the possibility that a cholinergic pathway maintains the baseline stiffness of body wall MCT through tonic activity that persists in isolated preparations. However, atropine can have direct non-receptor-mediated biochemical [70] and cytotoxic [71] effects on cells.

### 3.3. Extrinsic Stomach Retractor Apparatus

Apart from its involvement in autotomy, the ESRA is likely to have a passive role as a system of ligaments that limit displacement of the cardiac stomach wall and, through the presence of the contractile components of the retractor strands, contribute to the retraction of the stomach after extra-oral feeding. Although the relative importance of these two functions remains to be established, the contractile activity of the retractor strands (of *A. forbesi*) has been demonstrated experimentally using electrical stimulation and acetylcholine as excitatory agents [72,73]. Further confirmation of its contractile role emerged from recent work on the regulation of feeding behaviour in *A. rubens*. This showed that the neuropeptides ArSK/CCK1, which causes stomach contraction, and ArPPLNP2, a cardiac stomach relaxant, are expressed in the retractor strands [74,75].

There has been no attempt to identify the mechanism by which the ESRA separates from the stomach wall at autotomy or to determine if the collagenous tissue of the ESRA has mutable mechanical properties. If the ESRA normally has a restraining function and is, therefore, designed to resist tensile forces, it is highly likely that its rupture is facilitated by the destabilisation of its collagenous component at a potential breakage zone.

### 3.4. Comments on Physiological Aspects

Isolated preparations of the dorsolateral and ambulacral body wall that include their respective breakage zones represent tractable model systems for the in vitro investigation of the physiology of the arm detachment mechanism in *A. rubens* and other starfish. This methodology has shown that exposure to elevated K^+^ concentrations results in the nervously mediated weakening of dorsolateral and ambulacral preparations, culminating in rupture at their breakage zones. Such weakening can be attributed mainly to the collagenous dermis of the body wall, both (1) on the a priori basis that, being by far the most abundant structural material, it is primarily responsible for the mechanical integrity of the body wall, and (2) because of the morphological evidence for the disaggregation of dermal collagenous structures at the breakage zones during autotomy (see Section 2.2.2 and Section 2.3.2 above).

During autotomy undergone by intact animals and during in vitro experiments using isolated preparations, rupture was never observed to occur outside the localised breakage zone of the dorsolateral body wall or the diffuse breakage region of the ambulacral body wall. However, while this indicates that the collagenous dermis at other levels in the arm lacks the capacity for drastic, irreversible destabilisation, it can undergo reversible changes in tensile properties. This was demonstrated experimentally in distal dorsolateral preparations of *A. rubens* that did not include the breakage zone (see Section 3.1 above) [19], in whole detached arms of *A. forbesi* [76], and in distal ambulacral preparations of *Pycnopodia helianthoides* that did not include the breakage region [55]. It is not known if this variation in the capacity for tensile change of closely adjacent mutable collagenous components is due to (1) differences in their extracellular matrix, e.g., with respect to macromolecular composition and/or the nature and distribution of intermolecular bonds, (2) differences in the type or activity of the effector cells—the juxtaligamental cells—that modulate the tensile properties of their extracellular matrix, or (3) differences in the innervation of the tissues. No dissimilarities in the histological organisation of soft tissues at and outside the breakage zones were observed in *A. rubens* (during the course of the present authors’ work) or in *A. forbesi* [54].

Pharmacological investigations have failed to identify neurotransmitter chemicals that might be involved in the activation of dermal destabilisation and, thereby, autotomy in *A. rubens*. However, they have provided evidence that tonic activity in a neural pathway that includes muscarinic receptors maintains a baseline level of stiffness in the dermal collagenous tissue. This is a further indication of the importance of cholinergic regulation of effector systems in *A. rubens* [77]. It is intriguing that atropine, by itself, stiffens the mutable capsular ligament (catch apparatus) of the spine joint of a sea-urchin [78], which is the opposite of its effect on the dermal collagenous tissue of asteriid starfish and, assuming that this action is receptor-mediated, implies that a cholinergic pathway could be responsible for tonic inhibition of stiffening in this ligament. The discrepancy between the effect of atropine on starfish and sea-urchin preparations may be a reflection of the extreme antiquity of the split between the subphyla Asterozoa (Asteroidea + Ophiuroidea) and Echinozoa (Echinoidea + Holothuroidea), which occurred around 500 million years ago [79], and the consequent scope for divergence in the evolution of the regulatory mechanisms that modulate MCT tensility.

## 4. Final Remarks

Although there is a long history of scientific interest in the phenomenon of starfish arm autotomy [80], the mechanism by which arms are detached has been seldom addressed. King [10] suggested that the autotomy plane coincided with the weakest point in the ambulacral body wall of *A. rubens*. This was endorsed by Cuénot [12], who asserted that, in *A. rubens*, “l’autotomie est preparée par un locus minoris resistentiae” (the latter term being taken from the medical lexicon [81]). However, Anderson [54] stated that the autotomy plane is not a site of lesser resistance, since attempts to manually break the arms of *A. forbesi* did not result in separation at this level. He also commented that, in response to electrical stimulation of the isolated dorsal body wall, “softening and tearing of muscular and connective tissues occur along a predetermined line,” which is the first reported indication that a change in the mechanical state of the collagenous dermis might be implicated in the detachment mechanism. The work of the present authors on isolated body wall preparations of *A. rubens* (summarised in Section 3.1 and Section 3.2 above) confirmed that the dermis at the breakage zone consists of MCT with the capacity to undergo nervously mediated destabilisation of sufficient magnitude to permit the passive disconnection of the arm.

All investigated echinoderm autotomy processes have been found to depend on the endogenous loss of mechanical integrity of structurally critical mutable collagenous structures, which can be detected in vitro as an extreme reduction in their tensile strength, tensile stiffness, or viscosity [6]. By way of contrast, most autotomy-associated fracture in other phyla appears, on the basis of current evidence, to be achieved mainly by the application of muscular force [82,83,84,85,86,87], although the physiological weakening of collagenous tissue may contribute to the autotomy of nudibranch cerata and hydromedusan tentacles [88,89].

The molecular mechanism underpinning the irreversible destabilisation of MCT that occurs during echinoderm autotomy is unknown. As observed in *A. rubens*, there is no evidence that collagen fibrils undergo any change. It is hypothesised that, at autotomy, interfibrillar cohesion is interrupted by a chemical factor that is released from JLCs—possibly an enzyme or a specific inhibitor of interfibrillar linkage [6,26]. Autotomy-associated MCT disaggregation in different echinoderm classes is in some cases accompanied by changes in the JLC ultrastructure that are indicative of a regulated secretory mechanism [26].

As well as typifying the role of MCT in echinoderm autotomy-associated detachment mechanisms, the basal arm autotomy plane of *A. rubens* represents an ideal model system for addressing outstanding problems in MCT biology. Dermal collagenous components that traverse the autotomy plane can undergo drastic and irreversible destabilisation, whereas their counterparts at transverse levels outside the autotomy plane demonstrate only reversible changes in tensile properties. Therefore, this model system provides a scope for the comparison of (1) MCT at the autotomy plane before and after detachment, (2) MCT outside the autotomy plane in different (reversible) mechanical states, and (3) anatomically equivalent mutable collagenous structures at and outside the autotomy plane. As shown in this review, the model is amenable to in vitro pharmacological investigations using isolated preparations, an approach that is required to fully elucidate the neural circuits which regulate MCT tensility. More importantly, the model provides an opportunity for the application of comparative proteomic analysis and other “-omics” methods [90,91] which are aimed at the molecular profiling of different mechanical states and characterising the effector role of juxtaligamental cells. An ”-omics”-based investigative approach might also further illuminate the connection between autotomy and arm regeneration by identifying cellular and molecular adaptations at the autotomy plane that explain the tendency for echinoderm regeneration to proceed more efficiently and more successfully after autotomy than after other types of traumatic injury [48,92,93].

## Figures and Tables

**Figure 1 marinedrugs-21-00138-f001:**
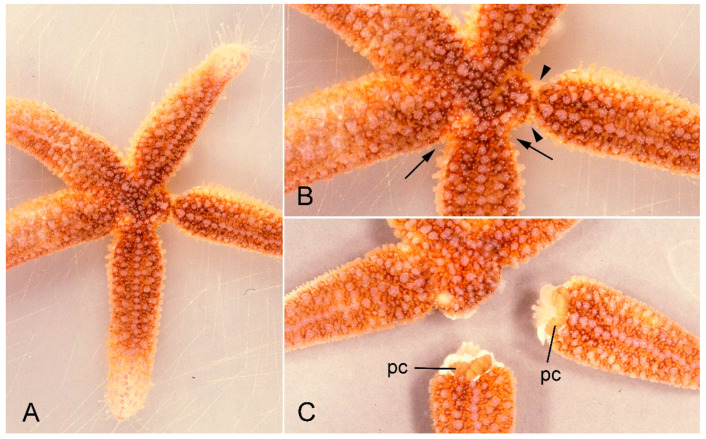
A specimen of *Asterias rubens* undergoing arm autotomy after intracoelomic injection of 0.56 M KCl. (**A**) General view of whole animal. (**B**) One arm shows basal constriction at the level of the autotomy plane (arrows) and the adjacent arm is undergoing detachment (arrowheads). (**C**) After complete detachment of both arms. pc, pyloric caeca.

**Figure 2 marinedrugs-21-00138-f002:**
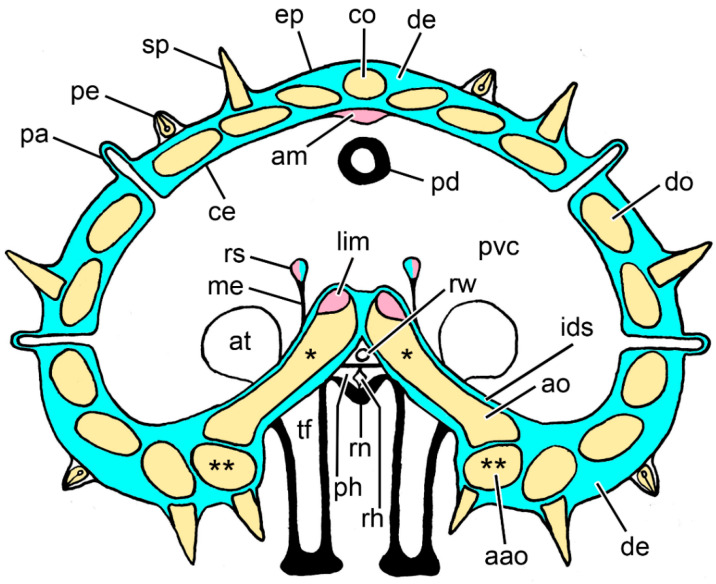
Diagram of a transverse section of an arm of *Asterias rubens* at the level of the autotomy plane. Not to scale. Blue: collagenous tissue; pink: muscle; yellow: ossicles; aao, adambulacral ossicle; am, apical muscle; ao, ambulacral ossicle; at, ampulla of tube-foot; ce, coelothelium; co, carinal ossicle; de, dermis; ep, epidermis; ids, inner dermal sheath; lim, longitudinal interambulacral muscle; me, mesentery; pa, papula; pd, pyloric duct; pe, pedicellaria; ph, perihaemal canal; pvc, perivisceral coelom; rh, radial haemal sinus; rn, radial nerve cord; rs, retractor strand; rw, radial water vascular canal; sp, spine; tf, tube-foot. * Adjacent pairs of ambulacral ossicles are connected by longitudinal interambulacral ligaments (not shown). ** Adjacent adambulacral ossicles are connected by longitudinal interadambulacral ligaments and muscles (not shown).

**Figure 3 marinedrugs-21-00138-f003:**
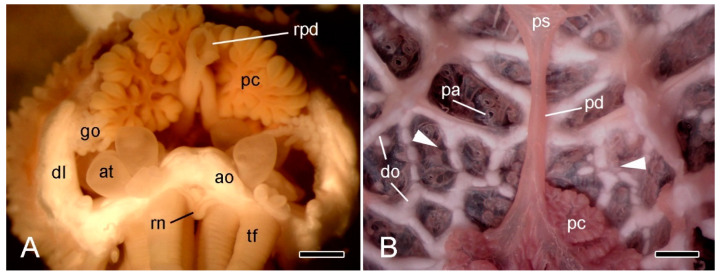
Stereomicroscopy (STM). (**A**) Proximal end of detached arm of *A. rubens* in frontal view immediately after autotomy. Scalebar = 1 mm. (**B**) Inner view of dorsal body wall at base of intact *A. rubens* preserved in 70% ethanol; distal end at bottom. Scalebar = 1 mm. Arrowheads, approximate position of breakage zone; ao, ambulacral ossicle; at, ampulla of tube-foot; dl, dorsolateral body wall; do, dorsolateral ossicles; go, gonad; pc, pyloric caecum; pd, pyloric duct; ps, pyloric stomach; rpd, ruptured end of pyloric duct; rn, radial nerve cord; tf, tube-foot.

**Figure 5 marinedrugs-21-00138-f005:**
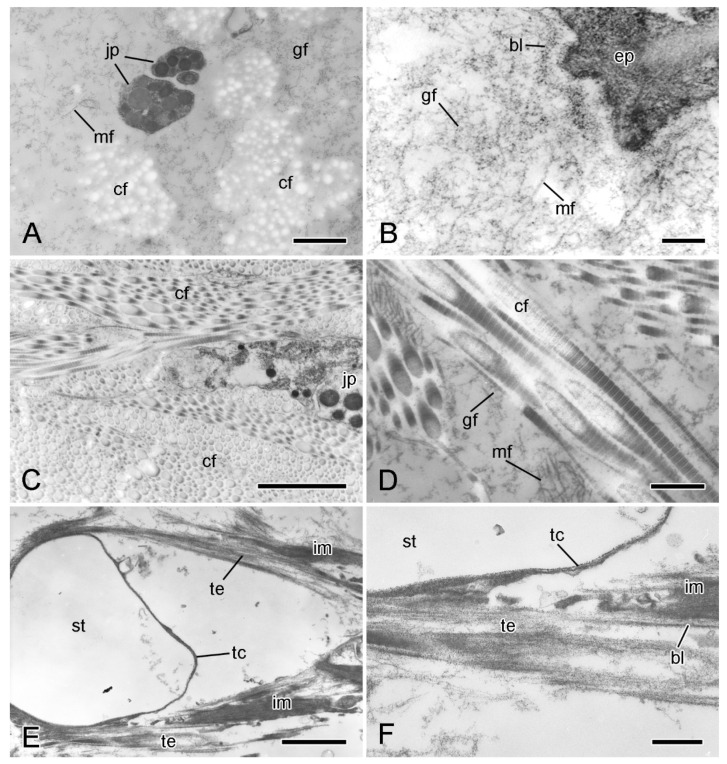
Transmission electron microscopy (TEM). Ultrastructure of dorsolateral body wall: extracellular components. (**A**) Expanded sparsely fibrous layer (ESFR) of outer dermis: general view. Transverse sections of collagen fibrils appear as electron-lucent spaces. Scalebar = 1 µm. (**B**) ESFR adjacent to epidermis. Scalebar = 0.2 µm. (**C**) Inner dermis: general view. Scalebar = 2 µm. (**D**) Inner dermis. Scalebar = 0.5 µm. (**E**,**F**) Tendons of interossicular muscle. (**E**) General view. Scalebar = 2 µm. (**F**) Junction between tendon and stereom trabecular coat. Scalebar = 0.5 µm. bl, basal lamina; cf, collagen fibril; ep, epidermis; gf, granulo-filamentous material; im, interossicular muscle; jp, juxtaligamental cell process; mf, microfibril; st, decalcified stereom bar; tc, trabecular coat; te, tendon ((**A**) adapted by permission of Taylor and Francis Group, LLC, a division of Informa plc., from reference [19], copyright 1990).

**Figure 6 marinedrugs-21-00138-f006:**
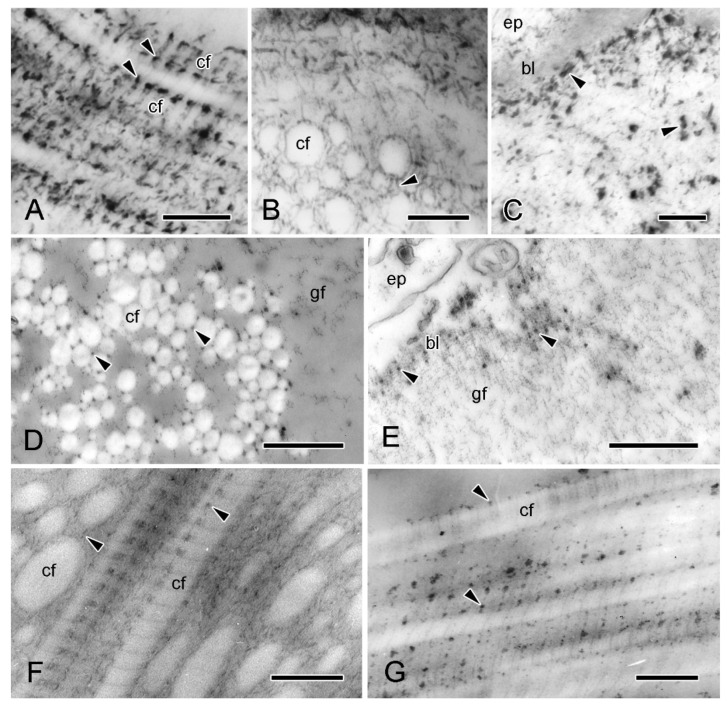
TEM. Proteoglycan histochemistry of dorsolateral body wall. (**A**–**E**) ESFR. (**A**–**C**) Cupromeronic blue (method of Erlinger et al. [37]). (**A**) Longitudinal section of collagen fibrils showing electron-dense deposits arranged regularly at each D-period (arrowheads). Scalebar = 0.2 µm. (**B**) Transverse sections of collagen fibrils with deposits appearing as interfibrillar bridges (arrowhead). Scalebar = 0.2 µm. (**C**) ESFR adjacent to epidermis, with deposits at basal lamina and in granulo-filamentous areas (arrowheads). Scalebar = 0.2 µm. (**D**,**E**) Polyethyleneimine (method of Sauren et al. [36]). (**D**) Electron-dense granules located on outer surface of collagen fibrils (arrowheads). Scalebar = 0.5 µm. (**E**) Adjacent to epidermis, basal lamina is labelled (arrowheads) but granulo-filamentous areas are unlabelled. Scalebar = 0.5 µm. (**F**,**G**) Inner dermis. (**F**) Cupromeronic blue: deposits arranged D-periodically on surface of collagen fibrils and forming interfibrillar bridges (arrowheads). Scalebar = 0.2 µm. (**G**) Polyethyleneimine: electron-dense granules arranged D-periodically. Scalebar = 0.2 µm. bl, basal lamina; cf, collagen fibril; ep, epidermis; gf, granulo-filamentous material.

**Figure 7 marinedrugs-21-00138-f007:**
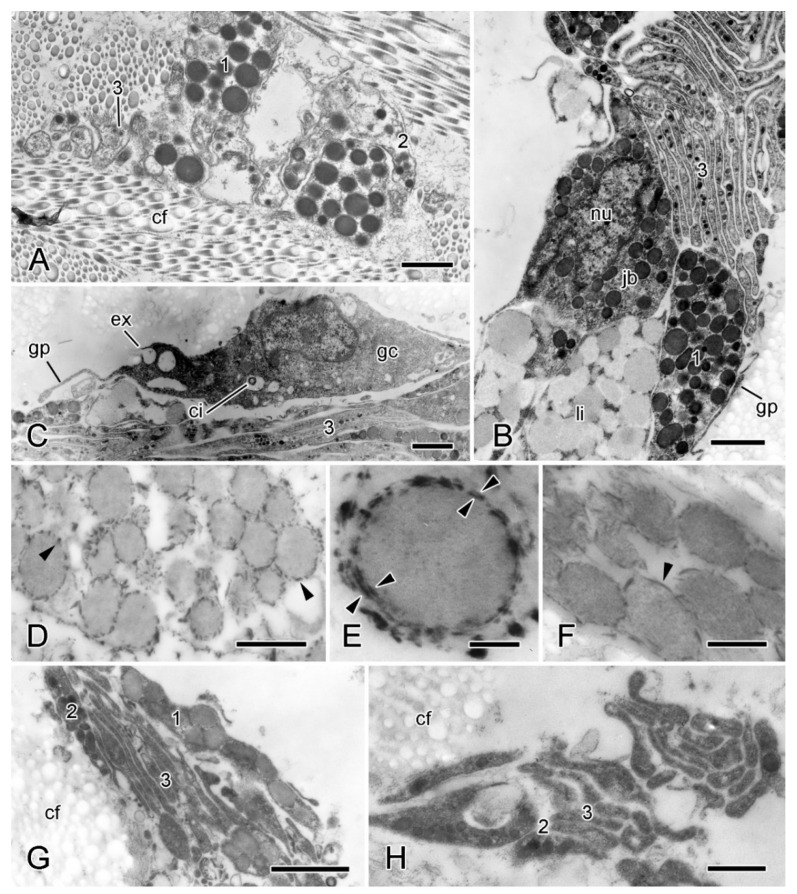
TEM. Ultrastructure of dorsolateral body wall: cellular components of inner dermis. (**A**–**F**) Intact body wall. (**A**) Three types of LDCV-containing JLC processes are present (1–3). Scalebar = 1 µm. (**B**) Aggregation of cell bodies and processes. Type 1 and 3 JLC processes and a nucleated type 1 juxtaligamental cell body are visible. Another cell body contains many presumptive lipid droplets. Scalebar = 1 µm. (**C**) Edge of cellular aggregation with possible gliocyte. Scalebar = 1 µm. (**D**–**F**) LDCVs of JLCs in sections stained with cupromeronic blue. (**D**) ESFR. Electron-dense deposits are present at LDCV membranes (arrowheads). Scalebar = 0.5 µm. (**E**) ESFR. Type 1 LDCV showing paired deposits separated by vesicle membrane (arrowheads). Scalebar = 0.1 µm. (**F**) Inner dermis. Electron-dense deposits are present at LDCV membranes (arrowhead). Scalebar = 0.2 µm. (**G**,**H**) JLC processes in autotomising body wall. (**G**) Type 1 LDCVs (1) show reduction in electron density, which is not evident in type 2 LDCVs (2). Scalebar = 1 µm. (**H**) Cluster of type 2 and 3 JLC processes alone. Scalebar = 0.5 µm. cf, collagen fibrils; ci, cilium; ex, vesicle undergoing exocytosis; jb, juxtaligamental cell body; gc, possible gliocyte; gp, possible gliocyte process; li, possible lipid inclusion; nu, nucleus ((**G**) and (**H**) adapted by permission of Taylor and Francis Group, LLC, a division of Informa plc., from reference [19], copyright 1990).

**Figure 8 marinedrugs-21-00138-f008:**
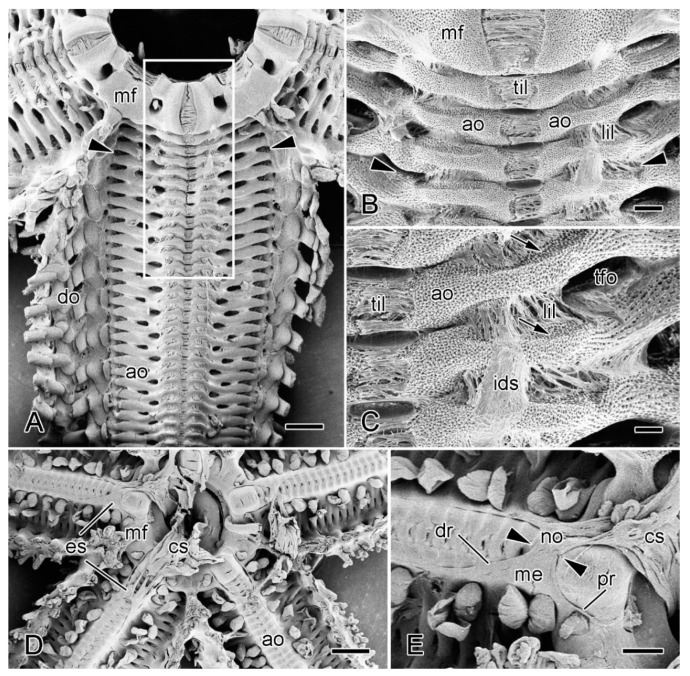
Scanning electron microscopy (SEM). Dorsal views of specimens of *A. rubens* from which most of dorsolateral body wall has been removed. (**A**–**C**) Soft tissues incompletely digested with bleach. (**A**) General view of one arm and adjacent mouth frame. Arrowheads indicate commonest level of breakage at autotomy (in larger animals). Box delineates margins of isolated preparations of ambulacral body wall used in physiological experiments (see Section 3.2). Scalebar = 1 mm. (**B**) More magnified view of proximal portion of specimen in A. Arrowheads indicate commonest level of breakage at autotomy. Scalebar = 0.2 mm. (**C**) Three interambulacral joints showing remnants of collagenous components. Arrows indicate attachment areas of longitudinal interambulacral muscles, which have been completely digested. Scalebar = 0.1 mm. (**D**,**E**) Undigested specimen showing location of extrinsic stomach retractor apparatus (ESRA). (**D**) General view. Scalebar = 1 mm. (**E**) ESRA associated with one arm. Arrowheads indicate location of autotomy breakage zone. Scalebar = 0.4 mm. ao, ambulacral ossicle; cs, cardiac stomach; do, dorsolateral ossicles; dr, distal retractor strand; es, extrinsic stomach retractor apparatus; ids, inner dermal sheath; lil, longitudinal interambulacral ligament; me, mesentery; mf, mouth frame; no, nodule; pr, proximal retractor strand; tfo, opening for tube-foot; til, transverse interambulacral ligament.

**Figure 9 marinedrugs-21-00138-f009:**
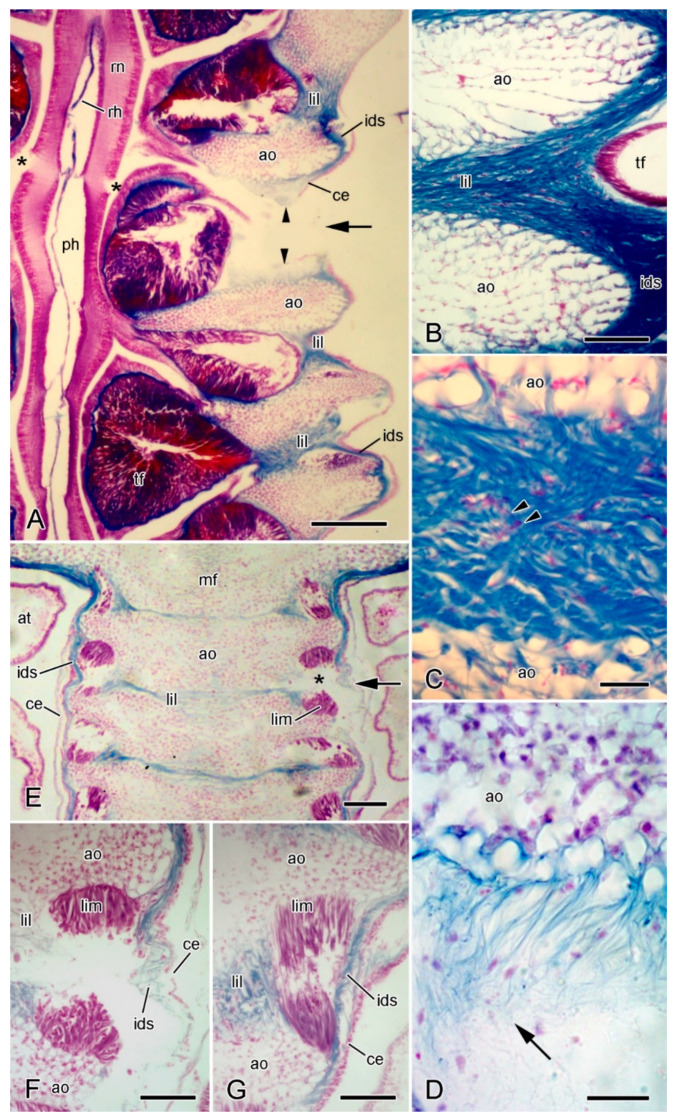
LM. Histology of ambulacral body wall. Horizontal sections stained with Milligan’s trichrome. In all sections, distal end is at bottom. (**A**) General view of one side of ambulacral ridge of arm undergoing autotomy, with radial nerve cord included. Arrow indicates breakage plane at which longitudinal ambulacral ligament has ruptured (arrowheads). Adjacent longitudinal interambulacral ligaments are intact. Radial nerve has started to fracture (asterisks). Scalebar = 0.2 mm. (**B**) Intact longitudinal interambulacral ligament, which is continuous with inner dermal sheath. Scalebar = 0.1 mm. (**C**) Intact longitudinal interambulacral ligament, which includes acidophilic cell bodies and processes (arrowheads). Scalebar = 20 µm. (**D**) Ruptured longitudinal interambulacral ligament at autotomising breakage plane, showing fibre disaggregation (arrow). Scalebar = 20 µm. (**E**–**G**) Ambulacral ridge of arm of another animal undergoing autotomy. Plane of section at more dorsal level than those shown in (**A**–**D**). (**E**) General view. Arm is undergoing autotomy at level indicated by arrow. Longitudinal interambulacral ligament at breakage plane is undergoing disaggregation. Longitudinal interambulacral muscles have ruptured at breakage plane (asterisk) and at other levels. Scalebar = 0.1 mm. (**F**) More magnified view of breakage plane shown in (**E**). Longitudinal interambulacral muscle has ruptured completely; longitudinal interambulacral ligament, inner dermal sheath and coelothelium have also ruptured. Scalebar = 50 µm. (**G**) Longitudinal interambulacral muscle and adjacent tissues at level two interambulacral joints distal to that shown in (**F**). Muscle rupture is less advanced than in (**F**). Longitudinal interambulacral ligament, inner dermal sheath and coelothelium appear to be intact. Scalebar = 50 µm. ao, ambulacral ossicle; at, ampulla; ce, coelothelium; ids, inner dermal sheath; lil, longitudinal interambulacral ligament; lim, longitudinal interambulacral muscle; mf, mouth frame; ph, perihaemal canal; rh, radial haemal sinus; rn, radial nerve cord; tf, tube-foot.

**Figure 10 marinedrugs-21-00138-f010:**
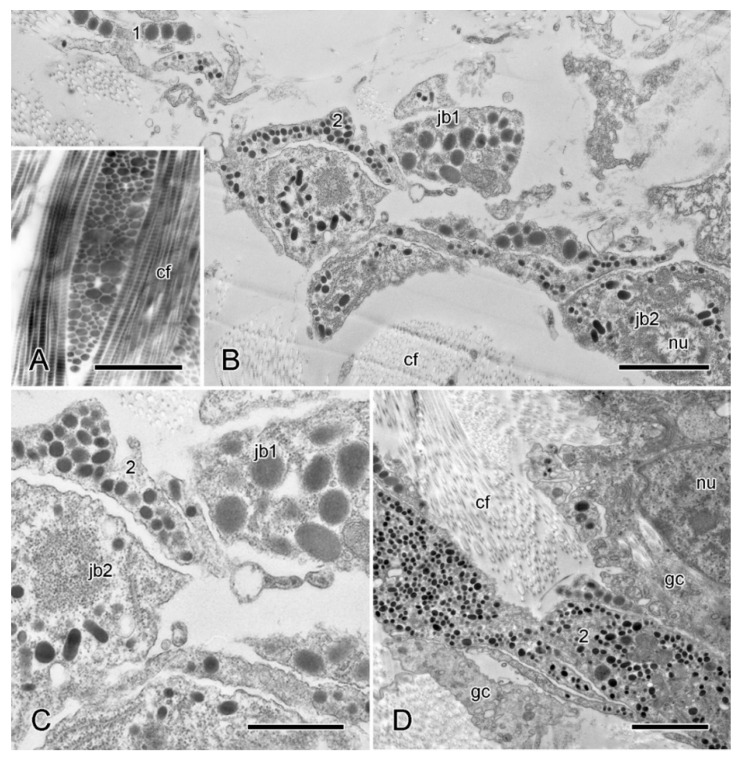
TEM. Ultrastructure of longitudinal interambulacral ligament. (**A**) Longitudinal and transverse sections of collagen fibres, each fibre comprising a tightly packed bundle of fibrils. Scalebar = 1 µm. (**B**) Cellular aggregation. Two types (1, 2) of JLC processes and somata are present. Scalebar = 2 µm. (**C**) Enlarged detail of JLC components in (B). Scalebar = 1 µm. (**D**) Cellular aggregation that includes presumptive gliocyte components adjacent to type 2 JLC processes. Scalebar = 2 µm. cf, collagen fibrils; gc, presumptive gliocyte; jb, juxtaligamental cell body; nu, nucleus.

**Figure 11 marinedrugs-21-00138-f011:**
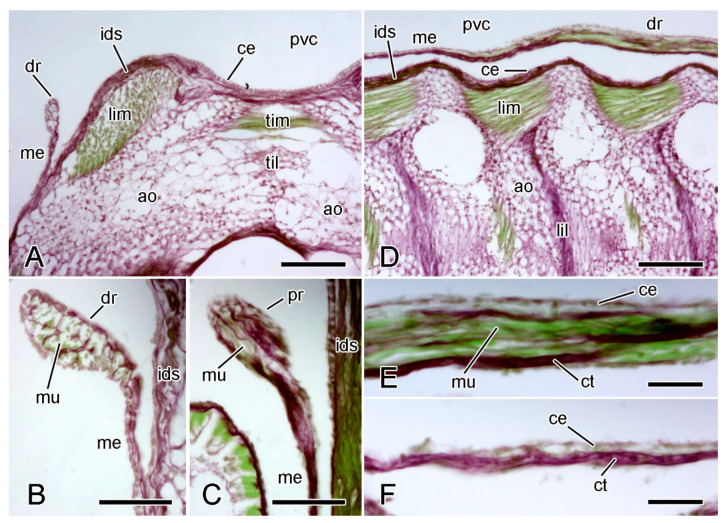
LM. Histology of extrinsic stomach retractor apparatus. Transverse (**A**–**C**) and horizontal (**D**–**F**) sections stained with Halami’s aldehyde fuchsin and light green [59]: collagenous tissue is purple, muscle and other cellular tissues green. (**A**) General view of dorsal region of one half of ambulacral ridge, including distal retractor strand. Scalebar = 0.1 mm. (**B**) Distal retractor strand and mesentery. Scalebar = 40 µm. (**C**) Proximal retractor strand and mesentery. Scalebar = 40 µm. (**D**) General view of one half of ambulacral ridge, including distal retractor strand. Scalebar = 0.1 mm. (**E**) Distal retractor strand. Scalebar = 20 µm. (**F**) Distal mesentery. Scalebar = 20 µm. ao, ambulacral ossicle; ce, coelothelium; ct, collagenous tissue; dr, distal retractor strand; ids, inner dermal sheath; lil, longitudinal interambulacral ligament; lim, longitudinal interambulacral muscle; me, mesentery; mu, muscle; pvc, perivisceral coelom; til, transverse interambulacral ligament; tim, transverse interambulacral muscle.

**Figure 12 marinedrugs-21-00138-f012:**
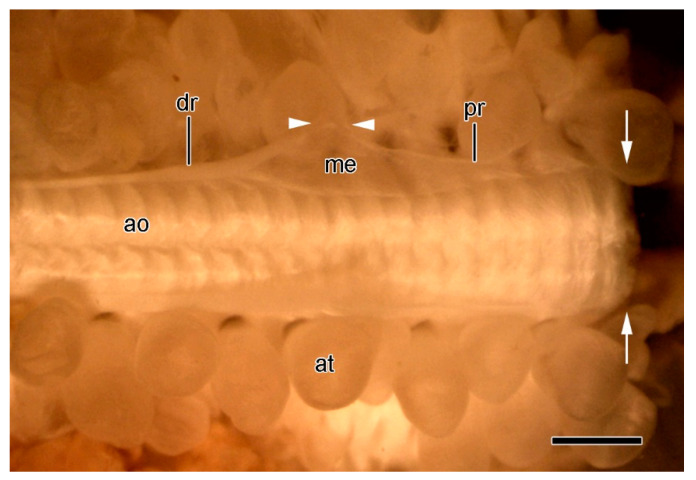
STM. Extrinsic stomach retractor apparatus (ESRA) immediately after autotomy. Dorsal view of ambulacral ridge; dorsolateral body wall removed. Distal end is on left. Arrows show location of ambulacral breakage plane. Components of ESRA on only one side are labelled. Arrowheads indicate location of ESRA breakage point. Scalebar = 1 mm. dr, distal retractor strand; ao, ambulacral ossicle; at, ampulla; me, mesentery; pr, proximal retractor strand.

**Figure 13 marinedrugs-21-00138-f013:**
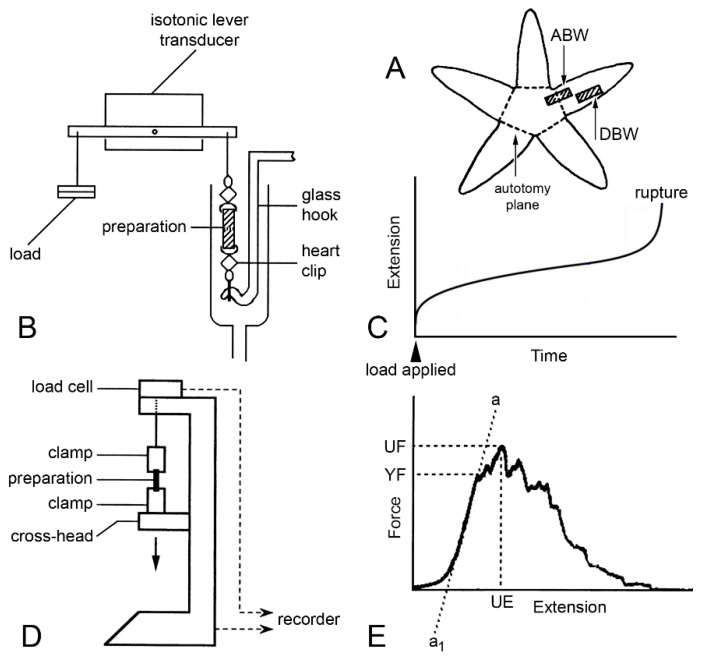
Mechanical tests. (**A**) Position of proximal dorsolateral body wall preparations that include autotomy plane (ABW) and distal preparations that lack it (DBW). (**B)** Diagram (not to scale) of apparatus used in creep tests in which preparations were subjected to constant load. (**C**) Idealised extension curve of ABW preparation under constant load. (**D**) Diagram (not to scale) of mechanical testing machine used for force-extension tests in which dorsolateral body wall preparations were stretched at a predetermined rate. (**E**) Force-extension curve showing measured parameters: ultimate extension (UE), from which ultimate strain was calculated (change in length/initial length), ultimate force (UF) from which ultimate stress was calculated (UF/cross-sectional area), yield force (YF) from which yield stress was calculated (YF/cross-sectional area), and slope of linear portion of curve (a…a_1_) from which Young’s modulus was calculated (Δstress/Δstrain, i.e., Δ(force/cross-sectional area)/Δ(extension/initial length)) ((**D**,**E**) adapted with permission of Springer Nature from reference [15], copyright 2000).

**Figure 14 marinedrugs-21-00138-f014:**
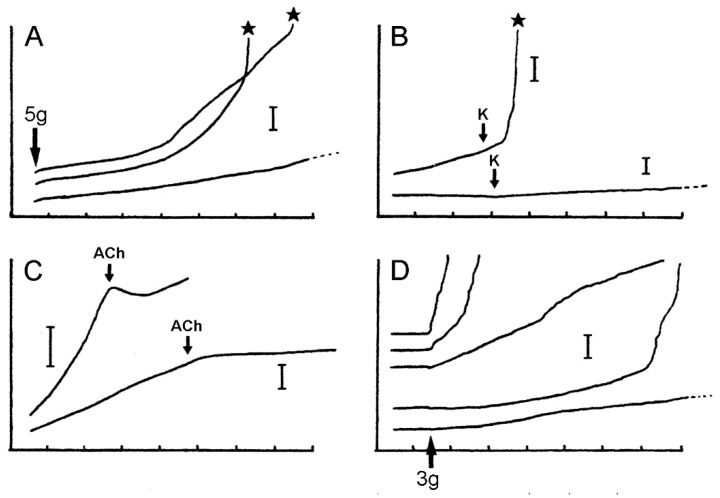
Extension curves of isolated dorsolateral body wall preparations subjected to constant load (see Figure 13B,C). Horizontal axis: time (min); vertical axis: extension, vertical bars indicating 1 mm; stars indicate rupture. (**A**) Normal extension curves of ABW preparations: three preparations from one animal. (**B**) Effect of 100 mM K^+^: typical response of ABW (upper curve) and DBW (lower curve) preparations. (**C**) Effect of 10^−3^ M acetylcholine on ABW preparations: usually acetylcholine caused sustained arrest of extension (lower curve); very rarely, it caused transient contraction (upper curve). (**D**) Effect of atropine alone: five ABW preparations from one animal which were immersed in either 10^−3^ M atropine (upper three curves) or sea-water alone (lower two curves) for 5 min before load was applied (adapted with permission of Taylor and Francis Group, LLC, a division of Informa plc., from reference [19], copyright 1990).

**Figure 15 marinedrugs-21-00138-f015:**
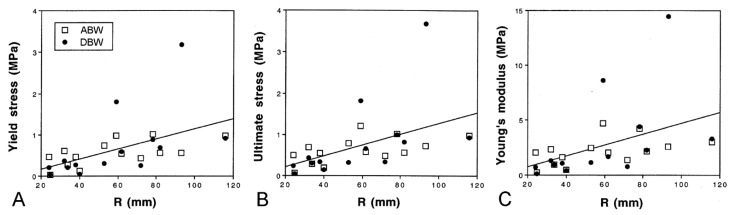
Size-related trends in mechanical properties of dorsolateral body wall preparations as demonstrated by force-extension tests (see Figure 13D,E). Regression lines (all slopes significant) for combined ABW and DBW preparations are included. (**A**) Yield stress. (**B**) Ultimate stress. (**C**) Young’s modulus. Parameters calculated from force-extension curves as explained in Figure 13E. R, centre to arm tip radius (adapted with permission of Springer Nature from reference [15], copyright 2000).

**Figure 16 marinedrugs-21-00138-f016:**
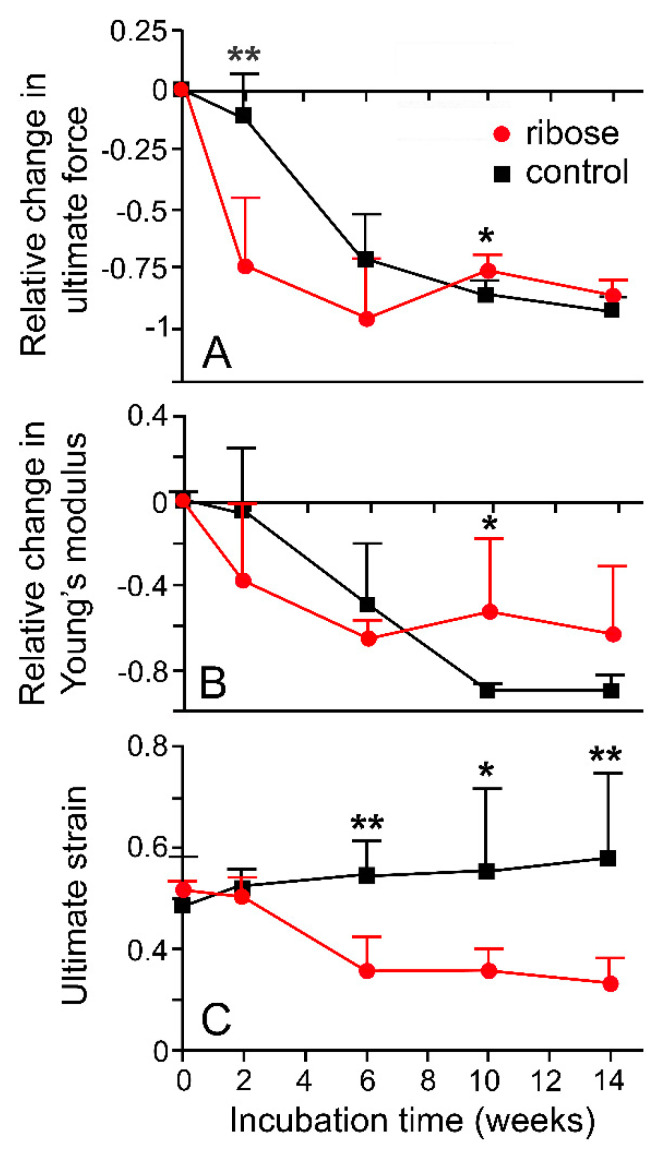
Effects of glycation on mechanical properties of dorsolateral body wall preparations. (**A**) Relative change in ultimate force (**B**) Relative change in Young’s modulus (**C**) Ultimate strain. DBW preparations were incubated in 0.2 M ribose in marine phosphate buffer (pH 7.6–7.8) at 30 °C; control preparations were incubated in buffer alone. Each data point is mean of six values; standard error bars are shown. ** *p* < 0.001; * *p* < 0.05 (one-way ANOVA and Bonferroni post-hoc test). See Figure 13E for explanation of parameters (I.C. Wilkie and J.J. Keane, unpublished data).

**Figure 17 marinedrugs-21-00138-f017:**
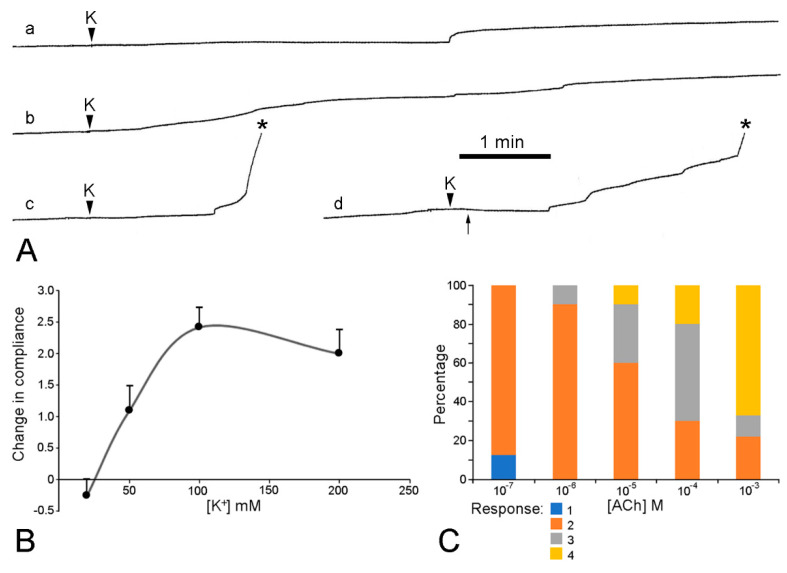
Effect of elevated [K^+^] and acetylcholine on ambulacral body wall preparations extending under constant load (see Figure 13B,C). Preparations consisted of central ambulacral ridge of basal region of arm and adjacent portion of mouth frame (Figure 8A). (**A**) Extension curves illustrating range of responses to 100 mM K^+^; a, transient increase in extension rate; b, sustained increase in extension rate; c, increase in extension rate culminating in rupture (asterisk); d, transient contraction (starting at arrow) then increase in extension rate and rupture. (**B**) Concentration-dependent effect of elevated [K^+^] on compliance of ambulacral body wall preparations. Responses, which varied qualitatively (see (**A**), a–d), were ranked in order of increasing magnitude of compliance change and scored (−1 to +4). Graph shows mean scores (*n* = 11–13 at each [K^+^]) and standard errors. (**C**) Concentration-dependent effect of acetylcholine on extension under constant load of ambulacral body wall preparations. Percentage of preparations (*n* = 9 or 10 at each [ACh]) exhibiting each type of response: 1, increased extension rate; 2, decreased extension rate; 3, extension arrested; 4, contraction. (I.C. Wilkie and G.V.R. Griffiths, unpublished data).

**Table 1 marinedrugs-21-00138-t001:** Pharmacology of ABW preparations. All agents were tested at a concentration of 10^−4^ to 10^−3^ M. Preparations were left for up to 10 min in 10^−3^ M solutions of cholinergic antagonists before either 10^−4^ M acetylcholine or 100 mM K^+^ was introduced. DMPP = 1,1-dimethyl-4-phenylpiperazinium; 0 = no significant effect. Data from Wilkie et al. (1990).

Agent	Effect
Acetylcholine	decrease in extension rate (+++)
Muscarinic agonists	contraction
methacholine	decrease in extension rate (++)
carbachol	decrease in extension rate (+)
pilocarpine	0
arecoline	0
Nicotinic agonists	
acetylthiocholine	0
nicotine	0
DMPP	0
Muscarinic antagonists	
atropine	increase in extension rate
	block of ACh effect
	no block of K^+^ effect
isopropamide	no block of ACh or K^+^ effects
Nicotinic antagonists	
hexamethonium	no block of ACh or K^+^ effects
tetraethylammonium	no block of ACh or K^+^ effects
Amines	
adrenaline	0
noradrenaline	0
dopamine	0
5-hydroxytryptamine	0
histamine	0

**Table 2 marinedrugs-21-00138-t002:** Pharmacology of ambulacral body wall preparations. All agents were tested at a concentration of 10^−3^ M. Preparations were left for 15 min in 10^−3^ M solutions of cholinergic antagonists and GABA agonists before either 10^−4^ M acetylcholine or 100 mM K^+^ was introduced. GABA = γ-aminobutyric acid; 0 = no significant effect. (I.C. Wilkie and G.V.R. Griffiths, unpublished data).

Agent	Effect
Acetylcholine	decrease in extension rate contraction
Muscarinic antagonists	
atropine	increase in extension rate
	block of ACh effect
	no block of K^+^ effect
isopropamide	no block of K^+^ effect
Nicotinic antagonists	
hexamethonium	no block of K^+^ effect
tetraethylammonium	no block of K^+^ effect
Amines	
tryptamine	0
5-hydroxytryptamine	0
octopamine	0
GABA agonists	
GABA	0
imidazole	0

## Data Availability

Not applicable.

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
