# Peer review of "Morphological and Physiological Aspects of Mutable Collagenous Tissue at the Autotomy Plane of the Starfish Asterias rubens L. (Echinodermata, Asteroidea): An Echinoderm Paradigm"

_marinedrugs, 2023, doi:10.3390/md21030138_

Round 1

Reviewer 1 Report

Comments and Suggestions for Authors

The manuscript by Wilkie and Carnevali represents an interesting body of research evidence about the starfish Asterias rubens L. and its mutable collagenous tissue, which involves in starfish autotomy. The manuscript represented good details about the autotomy mechanism of the starfish. However, the type of study is still ambiguous and it needs some revisions.

The authors stated that this study is a review study with some new data. However, most of the figures are new and they included new data in their selves. The authors must clarify which data is new. I highly recommend that this study rewrite in original article format and the result and discussion sections could be merged, and mention more details from previous literature about the results. 

If Figures 1-13 are original data, so where are the methods? The authors must declare the methods of the current study.

Non-original figures could be added in the supplementary file of the rewrote original manuscript.

Abstract 

The authors must add a brief method of the study in the Abstract section.

The names of species must be in italic form. Please unify the format in entire manuscript.

L 135-141. Where are the references? Are these sentences an original results of this study?

 The conclusion section must be brief, summarized of the whole study and accurate. The conclusion must be rewritten.

More updated references are needed.

Reviewer 2 Report

This is a well-written manuscript and a joy to review.

The list of references is complete and adequate. The quality of the images is superior. Tables follow the APA style and are well organized and displayed. The conclusions are well supported by the literature and evidence presented in the manuscript.

I detected only minor problems requiring the author's attention.

Some words need to be italicized, such as "A. rubens" in line 78 or "Asteria rubens" in line 95 (Figure 1's caption). That error is repeated in other places across the manuscript. Other species names also lack italics, for example, in line 808.

There seems to be an inconsistent usage of quotation marks. For example, quotes in lines 134 and 137 do not seem to match the style of the previous text, such as in line 70. The authors should double-check to be sure that they intended to use different quotation marks in those cases.

It appears that margins and indentation require some attention throughout the manuscript. For example, compare lines 330 to 343 with lines 344 and 345. Also, see line 525 for another example.

I have some additional minor suggestions.

Figure 16 could easily be made in greyscale.

I want the authors to double-check the list of abbreviations because I am under the impression that some may have been missed.

Reviewer 3 Report

I have no special comments on the content of the manuscript. I have only a small remark. In my opinion, species names are best written in italics, as well as Latin expressions, in particular, “in vitro”.

Reviewer 4 Report

The manuscript by Wilkie and Candia Carnevali is a nicely written review on arm autotomy in sea stars using Asterias rubens as a model species. The first part of the review describes the morphology and ultrastructure of the arm autotomy plane. Then, the effects of several pharmacological agents on the biomechanics of arm tissues are reviewed. Finally, these aspects are integrated in a model emphasizing the role of mutable collagenous tissue in autotomy. The text is clear and well-illustrated. I only have suggestions for minor modifications.

The layout of the text should be checked. Although it is unrelated with the quality of the manuscript, it interferes with the reading and understanding of the text. For example, there are parts of sentences hidden behind figures 9 and 10. The end of the paragraph after Table 1 is formatted as if it was part of the table, making it difficult to follow the text. Legend of Table 2 is split over two pages. Or, when there is a page break in the middle of a paragraph, the last line of the page and the first line of the following one should be justified.

Many species names throughout the text are not italicised.

In the morphological description of the dorsolateral body wall, the authors describe a space separating the outer and inner dermis. Could they describe more precisely the size or location of this space? I imagine it is not present everywhere otherwise there would be no mechanical connectivity between the two layers of the dermis.

Line 254, “LDVC” should be “LDCV”.

Lines 301-303 “The outermost layer…”, this sentence is not clear for me.

Line 306: “perineuronsl” ? “perineuronal”?

In the arm breaking plane, muscles apparently rupture through the middle by an unknown endogenous mechanism. Has such a similar already been described in other phyla in which autotomy occurs?

Line 560, “and a loose outer er of glia-like cells”? “outer layer”?

Lines 576-578 “the reduced electron opacity of type 1 LDCVs in apparently intact juxtaligamental processes seems unlikely to be a secondary effect and may signify the secretion of effector molecules implicated in dermal disaggregation”. Could the authors elaborate on this? In my view, if JLCs release the effector molecules by exocytosis, one could expect a decrease in the number of secretory granules but not necessarily a change in their appearance. Could the latter be related to an activation or maturation process?

In the manuscript, it is reported that the dorsolateral body wall distal to the autotomy plane is also a MCT able to reversibly switch from a soft to a stiff state. It therefore also encloses JLCs. Is there any information available about these cells (types, proportions, ultrastructure)? Are they different to those described in the breaking zone?

In the section about physiological aspects, I think there might be a mistake about the concentration of potassium showing an effect on MCT. Indeed, a concentration of 1 mM is reported several times, but this concentration is one order of magnitude lower than the natural concentration of potassium in seawater. Could the correct concentration be 100 mM? This would be in accordance with the data shown in figure 17B.

Lines 616-617 “the more usual sustained arrest of extension caused by acetylcholine represents passive stiffening of dermal MCT rather than myocyte contraction”. I do not understand the terminology “passive stiffening” in this case. If MCT if involved, it should be active?

Line 689 “This effect was usually transient and in some preparations was followed…”. I think the end of the sentence is missing here.

Round 2

Reviewer 1 Report

Based on the respected author final corrections the article is acceptable for publication.